# Fine-tuning of Geospatial Foundation Models for Aboveground Biomass Estimation

MICHAL MUSZYNSKI*, IBM Sustainability Software, Ireland

LEVENTE KLEIN, IBM Research - IBM TJ Watson Center, USA

ADEMIR FERREIRA DA SILVA, IBM Research Brazil, Brazil

ANJANI PRASAD ATLURI, IBM Sustainability Software, USA

CARLOS GOMES, IBM Research Europe, Switzerland

DANIELA SZWARCMAN, IBM Research Brazil, Brazil

GURKANWAR SINGH, IBM Sustainability Software, USA

KEWEN GU, IBM Sustainability Software, USA

MACIEL ZORTEA, IBM Research Brazil, Brazil

NAOMI SIMUMBA, IBM Research Japan, Japan

PAOLO FRACCARO, IBM Research Europe, UK

SHRADDHA SINGH, IBM Sustainability Software, USA

STEVE MELIKSETIAN, IBM Sustainability Software, USA

CAMPBELL WATSON, IBM Research - IBM TJ Watson Center, USA

DAIKI KIMURA, IBM Research Japan, Japan

HARINI SRINIVASAN, IBM Sustainability Software, USA

Global vegetation structure mapping is critical for understanding the global carbon cycle and maximizing the efficacy of nature-based carbon sequestration initiatives. Moreover, vegetation structure mapping can help reduce the impacts of climate change by, for example, guiding actions to improve water security, increase biodiversity and reduce flood risk. Global satellite measurements provide an important set of observations for monitoring and managing deforestation and degradation of existing forests, natural forest regeneration, reforestation, biodiversity restoration, and the implementation of sustainable agricultural practices. In this paper, we explore the effectiveness of fine-tuning of a geospatial foundation model to estimate Above-Ground Biomass (AGB) using space-borne data collected across different eco-regions in Brazil. The fine-tuned model architecture consisted of a Swin-B transformer as the encoder (i.e., backbone) and a single convolutional layer for the decoder head. All results were compared to a U-Net which was trained as the baseline model.

Experimental results of this sparse-label prediction task demonstrate that the fine-tuned geospatial foundation model with a frozen encoder has comparable performance to a U-Net trained from scratch. This is despite the fine-tuned model having 13 times less parameters requiring optimization, which saves both time and compute resources. Further, we explore the transfer-learning capabilities of the geospatial foundation models by fine-tuning on satellite imagery with sparse labels from different eco-regions in Brazil.

CCS Concepts: • **Applied computing** → **Environmental sciences**.

Additional Key Words and Phrases: aboveground biomass estimation, geospatial foundation models, fine-tuning

---

*the corresponding author for this research.

---

*KDD 2024, Aug. 25-29th, 2024*

© 2018 Association for Computing Machinery.

ACM ISBN 978-1-4503-XXXX-X/18/06...$15.00

https://doi.org/XXXXXXX.XXXXXXX

**ACM Reference Format:**
Michal Muszynski, Levente Klein, Ademir Ferreira da Silva, Anjani Prasad Atluri, Carlos Gomes, Daniela Szwarcman, Gurkanwar Singh, Kewen Gu, Maciel Zortea, Naomi Simumba, Paolo Fraccaro, Shraddha Singh, Steve Meliksetian, Campbell Watson, Daiki Kimura, and Harini Srinivasan. 2024. Fine-tuning of Geospatial Foundation Models for Aboveground Biomass Estimation. In *Proceedings of the KDD 2024 conference - Fragile Earth Workshop, August 25-29th, 2024, Barcelona, Spain.* ACM, New York, NY, USA, 8 pages. https://doi.org/XXXXXXX.XXXXXXX

## 1 INTRODUCTION

The accurate estimation of forest attributes, such as tree height, plays an important role in understanding forest structure and changes in biomass and carbon sequestration. Forest services typically use traditional field-based methods to compile forest inventory data, examining tree attributes such as tree height, diameter at breast height, canopy diameter, and species type. These measurements are then converted into tree biomass using species-specific alometric equations [11], which can then be used to estimate the carbon sequestrated in each tree. However, field-based measurements are time-consuming, labor-intensive, expensive, and limited to accessible locations, thus making the compiled datasets sparse in space and time and difficult to generalize to large scales [8].

Remote sensing techniques have emerged as a promising alternative to traditional in-situ measurement methods, offering cost-effective solutions for tree height estimation at various spatial and temporal scales. Among the different remote sensing methods of relevance to vegetation structure mapping, airborne and spaceborne Light Detection and Ranging (LiDAR) systems, Synthetic Aperture Radar (SAR), and multispectral and hyperspectral imagery have gained significant attention in recent years [25].

LiDAR technology has proven highly effective in estimating canopy height and delineating trees, offering a high degree of accuracy [8]. Although high-quality LiDAR data, such as aerial LiDAR, has been gathered in numerous locations and its significance acknowledged, the availability of such data is not uniform worldwide. However, the advent of space-based LiDAR observations, such as the Global Ecosystem Dynamics Investigation (GEDI) [6] and the Ice, Cloud, and Land Elevation Satellite (ICESAT-2) [17], has provided direct measurements of vegetation characteristics, including canopy height, in regions previously unmeasured. Nonetheless, these space-based LiDAR measurements are sparse in terms of spatial distribution. Consequently, the process of transforming these sparse data points into spatially continuous AGB estimates requires the use of geostatistical methods including machine learning [1, 12].

Predicting forest characteristics such as Above-Ground Biomass (AGB) and canopy height has been explored using supervised machine learning approaches. Examples include the use of linear regression [1], ensemble methods such as random forests [7] and gradient boosting [13], support vector machines [23], and more recently the use of deep learning approaches based on deep neural networks [12, 16, 19].

A recent self-supervised learning breakthrough in artificial intelligence are models known as foundational models. These are large models that learn global patterns and general features from extensive unlabeled data. In comparison, classical deep learning models such as Convolutional Neural Networks (CNNs) and U-Nets focus training on local dependencies only [10]. The foundation model approach, that is very successful in language modeling, has been extended to images where masked auto-encoders partially obscure images for the reconstruction of masked parts [9]. Satellite imagery with its abundant and open availability of moderate spatial resolution Landsat and Sentinel satellites have been used to successfully build Geospatial Foundation Models (GFMs).

The main obstacle to fully benefit from automated general-purpose computer vision tools for geospatial applications is a shortage of very-large-scale, multi-task remote sensing datasets. Hence,

there is a keen interest in self-supervised methods which can gain general domain knowledge from unlabelled datasets. For example, state-of-the-art computer vision architectures, such as Contrastive Language-Image Pre-Training (CLIP) models [21], Vision Transformers [5] and Swin Transformers [14] can be pre-trained on large datasets with an aim to not over-fit. Labels are then required for fine-tuning with tasks ranging from object detection, instance segmentation, and semantic segmentation. A limited number of labelled remote sensing datasets presently exist such as the DOTA dataset [27], the iSAID dataset [26], and the Deep-Globe dataset [4]. These datasets contain less than 10K images, though this is significantly less than the millions of images used to train the CLIP.

In this paper, we aim to test geospatial foundation models for predicting AGB from NASA's HLS imagery. The main contributions of this paper are the following:

- We investigate if fine-tuning of geospatial foundation models with a frozen encoder and only 0.6 million tunable parameters can match the capabilities of a state of the art U-Net with 7.8 million tunable parameters.
- We investigate the generalizability of these foundation models across different eco-regions in Brazil.

## 2 RELATED WORK

### 2.1 Self-Supervised and Multi-Task Learning for Remote Sensing.

Supervised learning is often used for feature extraction from labeled data, but it requires large amounts of labeled data for model training. This can be challenging, especially in the context of remote sensing, where manual annotation of large datasets is often impractical. Additionally, the location sensitivity of annotations such as variations in acquisition geometry, atmospheric conditions, and land cover phenology can affect the transferability of trained models to new areas. To address the need for manual annotation, Self-Supervised Learning (SSL) has been explored as a method for training models using large sets of unlabeled data. In the context of deep learning, particularly with images, SSL typically involves two tasks: a self-supervised pretext task for model training, where unlabeled data is manipulated to generate pseudo-labels, and real downstream tasks associated with applications. The success of SSL depends heavily on the design of the pretext task [30]. A well-designed pretext task helps the network capture high-level representations of the input data, enabling the model to learn from a large volume of unlabeled data. Two common strategies for pretext design are the generative-based tasks that reconstruct parts of the intentionally perturbed input data, and contrastive-learning based tasks which differentiate inputs with similar meanings [30]. Those pre-training of SSL seems to the most appreciate approach to deal with Earth observation tasks with sparse labels, for example, GEDI measurements.

Different approaches have been developed to improve performance on a wide range of downstream applications with very few labels. Several proposed methods incorporate temporal augmentations into a contrastive learning framework in which image tiles of the same location captured at different times are imposed to have more similar representations than images of different locations [15]. Furthermore, a general-purpose neural architecture with a focus on geospatial tasks was proposed by combining self-supervised learning with supervised training on diverse tasks [22]. The state of the art computer vision models cannot handle all the label types that naturally occur in different geospatial downstream tasks. For example, Mask2Former [2] can simultaneously perform semantic and instance segmentation, though it is not capable of predicting properties of polygons or classify images due to its architecture design. That does not allow these models to benefit from transfer learning opportunities in the field of Earth observations. For example, pre-trained models

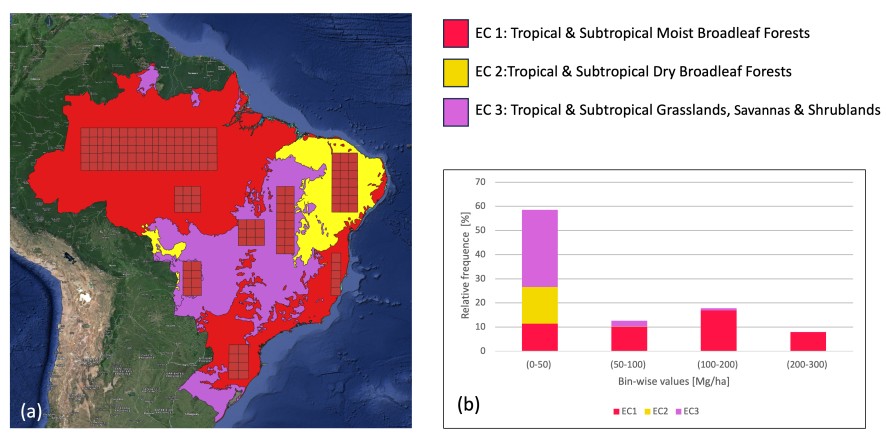

Fig. 1. Overview of the Brazil 2022 dataset: a) image tiles selected from three eco-regions in Brazil: the eco-region Tropical and Subtropical Moist Broadleaf Forests (EC1), the eco-region Tropical and Subtropical Dry Broadleaf Forests (EC2) and the eco-region Tropical and Subtropical Grasslands, Savannas and Shrublands (EC3) to build the dataset and b) the distribution of GEDI measurement values for the image tiles from each of the three eco-regions.

on detecting building polygons could improve image segmentation for land cover and land use, since land use includes a human-developed category.

There are still many unanswered questions regarding how to effectively adapt methods commonly used in other domains to the specific properties of Earth observation data. In particular, the potential of self-supervised learning in the context of above-ground biomass prediction has received limited attention, despite some promising results that have been reported [20].

## 3 DATASET

In the last decades, forests in Brazil have been subjected of de-forestation and conversion to agricultural lands. Being willing to explore biological changes in the Brazilian eco-system, we consider four main eco-regions in Brazil: Tropical and Subtropical Moist Broadleaf Forests (EC1), Tropical and Subtropical Dry Broadleaf Forests (EC2), Tropical and Subtropical Grasslands, Savannas and Shrublands (EC3), and Flooded Grasslands and Savannas (EC4) [18] in this work. We have merged EC3 with EC4 due to the small size of the eco-region EC4 and then we call the joint eco-region EC3. In Figure 1a), we show the distribution of GEDI measurements over those three main eco-regions collected in 2022. The majority of aboveground biomass measurements are in the bin (0-50) and the bin (100-200), as shown in Figure 1b), therefore our AGB regressors should mainly focus on accurate predictions of low and medium AGB values.

Preprocessing of raw satellite images is required and encompasses a range of additional steps. We describe the most relevant steps, below. Harmonized Landsat-8 Sentinel-2 (HLS) data that consist of six channels: Blue, Green, Red, NIR-Narrow, SWIR 1 and SWIR 2 are often contaminated with large amounts of clouds or no-data values. To ensure high quality data for training or fine-tuning, we proposed a pre-processing methodology that excludes images with large numbers of missing values and/or containing cloud coverage. To achieve that, we took advantage of the cloud mask corresponding to each HLS tile for a given timestamp. Across a time interval and leaf on season, all tiles are analyzed at pixel level and the median cloud free pixel is considered to create a cloud free image for a given area. The cloud free image is used to fine-tune the model for AGB prediction.

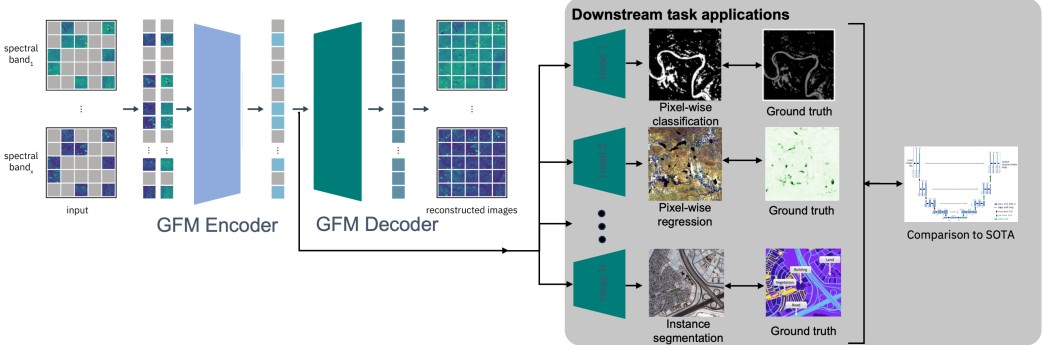

Fig. 2. The encoder-decoder architecture diagram of the geospatial foundational model used to estimate AGB in Brazil.

## 3.1 Fine-tuning, Validation, and Testing Set

To evaluate generalization of fine-tuned geospatial foundation models, we estimate the average performance over fine-tuning sets extracted from three major eco-regions in Brazil in 2022. All labeled data from a validation set (i.e., unseen data) is held out for an objective evaluation of fine-tuned geospatial foundation model regressors and our baseline, U-Net on samples from the same eco-region on which they have not been fine-tuned or trained.

## 4 METHODOLOGY

To address our AGB pixel-wise regression task, we fine-tune a geospatial foundation model that is based upon Prithvi [10] but with a Swin-B backbone and a state-of-the-art U-Net regressor. The models are fine-tuned and trained, respectively, on different training datasets representing three major eco-regions in Brazil. The details of the models are described in this section.

## 4.1 Fine-tuning the geospatial foundation model and training the U-Net

The foundation model is based upon that described in [10] except that the backbone is a Swin-B transformer [14]. Briefly, for pre-training, we leverage SimMIM [29], a SSL strategy based on masking large parts of the input and tasking the model with reconstructing them. Following SimMIM [29], during pre-training, a small decoder composed of a single convolutional layer followed by a Pixel Shuffle [24] module is used to reconstruct all image patches. In this work, we used two versions of the geospatial foundation model, one pre-trained with 1000 HLS image tiles sampled across the globe (i.e., Global GFM) and another pre-trained on HLS image tiles sampled across the US, mainly in Texas and Louisiana (i.e., Local GFM). More details can be found in [10].

For fine-tuning, we froze the encoders and replaced the decoder used in pretraining with a UPerNet [28] decoder, as suggested in [14], adapted for the pixel-wise regression task. The standard UPerNet implementation (available in mmseg [3]) using Swin-B as a backbone predicts a final feature map 4x smaller than the input. This is then be upsampled, typically through bilinear interpolation, to match the input size, before an `argmax` operation that is commonly used for pixel-wise classification. While such an operation may be reasonable in semantic segmentation, when one is limited to a small discrete set of classes, we find that it is unsuitable for regression tasks, producing blurry results similar to what is observed when bilinear interpolation is used on standard images. In order to resolve this, we append two Pixel Shuffle [24] layers to the UPerNet

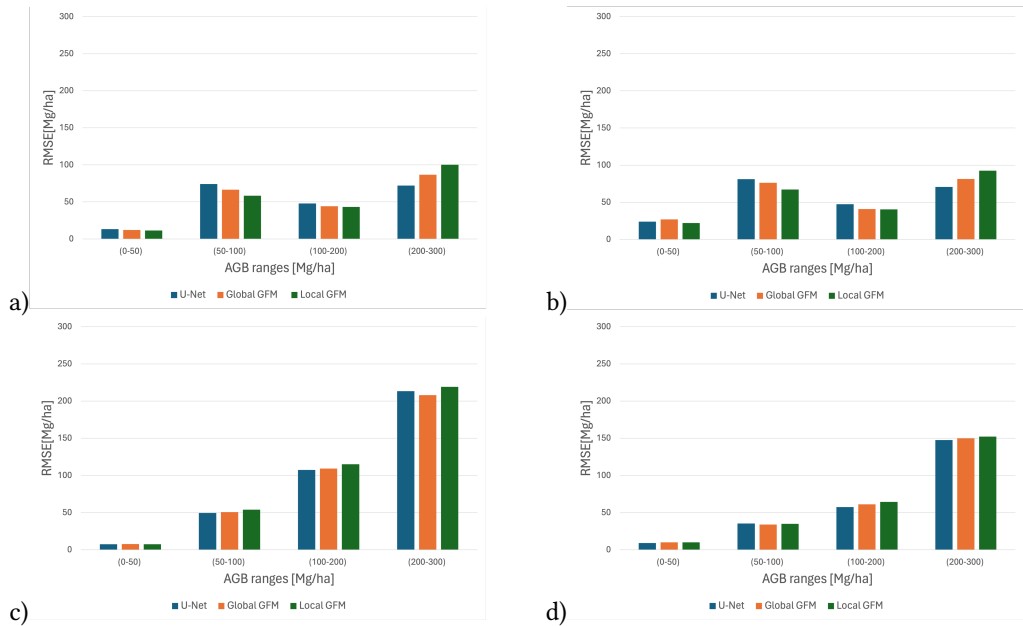

Fig. 3. Performance of AGB prediction models measured by bin-wise RMSE values on image tiles from: a) all the three eco-regions together, b) the eco-region Tropical and Subtropical Moist Broadleaf Forests (EC1), c) the eco-region Tropical and Subtropical Dry Broadleaf Forests (EC2) and d) the eco-region Tropical and Subtropical Grasslands, Savannas and Shrublands (EC3).

decoder, resulting in a learned 4x upscaling. The final adaptation of for the regression task is the prediction of a single output value and the introduction of a ReLU activation function.

The Global GFM and Local GFM were fine-tuned to estimate AGB using 8 A100 GPUs for 100 epochs, with a maximum learning rate of 2e-4 and a cosine decay schedule with a warm-up of 10 epochs.

For a baseline model, we use a U-Net-based architecture following the state-of-the-art work on for carbon storage and above-ground biomass estimation [19]. Considering the existing U-Net models, we selected a learning rate of 0.01 and a batch size of 128, that we consistently used across training of all U-Net based AGB regressors. We also used the Adam optimizer that has been proven to outperform classical optimizers in a range of scenarios to optimize our RMSE loss function. It is worth mentioning that fine-tuning of GFMs with frozen encoders requires optimization of only around 0.6 millions decoder parameters while training from scratch of U-Net involves learning of around 7.8 millions parameters.

Finally, we evaluated the performance of all regression models by calculating the bin-wise Root Mean Square Error (RMSE) on a validation set. It is important to mention that we remove pixels corresponding to the invalid values from our evaluation procedure, as no labels exist for those areas. Moreover, we provide statistics on model performance per Brazilian eco-region.

## 5 RESULTS

In Figure 3, we present the AGB prediction results of the Global GFM, Local GFM and U-Net regressor for image tiles from a) all the three eco-regions together, b) the eco-region EC1, c) the eco-region EC2 and d) the eco-region EC3. The bin-wise Root Mean Squared Error (RMSE) is

calculated only on the validation set. There are some differences in model performance across AGB ranges and ecoregions. For example, when we analyse the results for all three eco-regions together or EC1, the GFMs are more accurate across AGB ranges of 50-200 Mg/ha than the U-Net. However, overall the U-Net slightly outperforms the GFMs (i.e., Global GFM and Local GFM) by achieving an RMSE of 65.5 Mg/ha compared to 68.7 Mg/ha and 70.9 Mg/ha, respectively.

The performance of the GFMs is impressive given the tunable parameters amount to less than 10 percent compared to the U-Net. This results in a model that is faster to train and likely more robust to label-limited problems, common in forest-based applications.

To examine generalizability, we fine-tine those models on each of Brazilian eco-regions. For low AGB values (i.e., bin (0-50 Mg/ha)), we can observe that the Global GFM and Local GFM slightly outperform our baseline, U-net independently of eco-regions in Figure 3. However, we can see that U-Net predictions are characterized by lower RMSE for the eco-region EC2 and EC3, when we analyze moderate AGB values (i.e., bin (100-200 Mg/ha)).

## 6 CONCLUSIONS AND FUTURE WORK

In this work, we have investigated AGB predictions to estimate the total carbon sequestered in forests across the three different ecoregions in Brazil. We provided insights on the fine-tuning process of GFMs as well as a comprehensive evaluation of a task where labels are sparse. In terms of performance, we showed that the fine-tuned GFMs with frozen encoders match the performance of a state-of-the-art U-Net trained from scratch while having 13 times less parameters requiring optimization. Extending the transfer learning capabilities of geospatial foundation models to infer AGB in regions where data is sparse can provide quick insight into carbon sequestration. Future development of GFMs will include integrating new multimodal data sources and testing those models under more realistic conditions, for example, using radar in the presence of persistent cloud cover.

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
