# OpenReview forum: "Fine-tuning of Geospatial Foundation Models for Aboveground Biomass Estimation"
_KDD.org/2024/Workshop/Fragile_Earth — Fragile Earth FullPresentation_

### Official Review · Reviewer_JptZ · 2024-07-12
**Nicely written paper on GFM for Aboveground Biomass Estimation**

**Rating:** 7
**Confidence:** 4

**Review:**

This paper uses a fine tuned Swin-B transformer to estimate the biomass using using remote sensing imagery. Authors compare their results with U-Net segmentation architecture and illustrate the the proposed approach requires less computational resources. The proposed approach uses a pre-trained Swin-B as its encoder, and adds a convolutional layer as decoder to infer these. The experiments are conducted on NASA HLS imagery and validates if these foundational models are general enough for using for different regions. The paper looks to be technically sound, and the experiments are backing the claims by the authors. I have some question which can be addressed by the authors going forward. First, the results show that there is some RMSE differences for different regions, parameters. This require more explanation. While I anticipate locally trained model to perform better, I see in Figure 3, it is not always the case. I know the local is using some US regions, but still the intuitive reasoning behind these would help. Results section can go deeper into these details a little bit more. In addition, RMSE is a good indicator of the performance if the readers are from the same domain, but since I am not, I don't know how to interpret these, i.e. is this a good level of errors? Finally, the decoder is oversimplified, i.e. can be more carefully designed for this task. Maybe a set of experiments on this can help readers understand the impact of different decoder design decisions. In summary, despite some of the details needed, I see that the paper is well written and illustrates how state of the art models can be used on earth related problems. Therefore, I will be happy to see the paper presented in the workshop.

---

### Official Review · Reviewer_D9FS · 2024-07-13
**A solid paper on an important problem both from application (sustainability) and technical perspectives**

**Rating:** 7
**Confidence:** 4

**Review:**

This paper addresses the question of whether a fine-tuned geospatial foundation model can be leveraged to provide improved above-ground biomass estimation, which is critical for monitoring and managing deforestation, reforestation and other important metrics towards sustainability. Specifically, they evaluate the performance of a fine-tuned model with a Swim-B transformer and a frozen encoder having a magnitude less parameters as the state-of-the-art U-Net and show that it can nearly match the performance of the latter.
As a comment, it would be interesting to see what insights the authors may have on the variation in the relative performance between the proposed scheme and the baseline U-Net.
Overall, the paper addresses an important problem for the sustainability agenda (above ground biomass estimation) with an approach that is topical and its contents will benefit the workshop goals and audience.

---

### Official Review · Reviewer_9hnS · 2024-07-13
**Review of Fine-tuning of Geospatial Foundation Models for Aboveground Biomass Estimation**

**Rating:** 7
**Confidence:** 4

**Review:**

Summary:

This paper studies the effectiveness of fine-tuning of a geospatial foundation model to estimate above-ground biomass (AGB) using space-borne data collected across different eco-regions in Brazil. The fine-tuned model architecture consisted of a Swin-B transformer as the encoder (i.e., backbone) and a single convolutional layer for the decoder head. Experiments showcase that the fine-tuned geospatial foundation model with a frozen encoder achieves promising performance on real-world data.

Strengths:

+ This paper studies an important and interesting problem.
+ The paper is well-written and the proposed approach is easy to understand.
+ Performance of the proposed model in practice is clear and promising.

Weaknesses:
- It would be good if the authors can provide other evaluation metrics for comparison instead of only bin-wise Root Mean Square Error.
- It is not clear what are tables/boxes in Figure 1.

---

### Decision · Program_Chairs · 2024-07-24

Accept (Full Presentation)